# Fast and Scalable Inference of Dynamical Systems via Integral Matching

**Baptiste T. Rossi**
MIT
Cambridge, MA
brossi@mit.edu

**Dimitris J. Bertsimas**
MIT
Cambridge, MA
dbertsim@mit.edu

## Abstract

We present a novel approach to identifying parameters of nonlinear Ordinary Differential Equations (ODEs). This method, which is based on collocation methods, enables the direct identification of parameters from time series data by matching the integral of the dynamic with an interpolation of the trajectory. This method is distinct from existing literature in that it does not require ODE solvers or an estimate of the time derivative. Furthermore, batching strategies, by time subintervals and component of the state, are proposed to improve scalability, thus providing a fast and highly parallel method to evaluate gradients, and a faster convergence than adjoint methods. The effectiveness of the method is demonstrated on chaotic systems, with speed-ups of three orders of magnitude compared to adjoint methods, and its robustness to observational noise and data availability is assessed.

## 1 Introduction

This paper introduces a new approach for learning coefficients of ordinary differential equations from time series data. We focus on scalability and universality to learn possibly high-dimensional systems from large datasets while making minimal assumptions on the equations. The resulting algorithm can be used to infer parameters of equations derived from scientific laws or, when no prior structure is known, to train universal approximators such as the Neural ODEs from Chen et al. [2018].

### 1.1 The problem

Let $\mathbf{x}(t) \in \mathbb{R}^n$ be the state of a system of dimension $n$ at time $t$. We have $M$ observations $\mathbf{x}(t_m)$ over a single trajectory, where $t_m \in [0, T]$ for all $m = 1 \dots M$, and are interested in the inverse problem of inferring parameters $\mathbf{\Theta}^*$ and an initial condition $\mathbf{X}_0^*$ that minimize the prediction error on $\hat{\mathbf{x}}(t)$:

$$\min_{\mathbf{X}_0, \mathbf{\Theta}} \quad \frac{1}{M} \sum_{m \in [M]} \|\hat{\mathbf{x}}(t_m) - \mathbf{x}(t_m)\|^2$$

$$\text{s.t.} \quad \dot{\hat{\mathbf{x}}}(t) = f(\hat{\mathbf{x}}(t), \mathbf{\Theta}), \quad \forall t, \quad \text{(1a)}$$

$$\hat{\mathbf{x}}(0) = \mathbf{X}_0. \quad \text{(1b)}$$

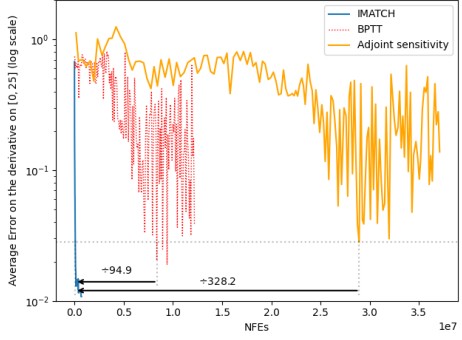

Figure 1: On the damped oscillator from Chen et al. [2018], our algorithm (blue) fits a neural ODE with fewer network evaluations and to greater accuracy than Backpropagation through time (BPTT) (red) and Adjoint sensitivity (orange). The best accuracy reached by the adjoint sensitivity within 15 minutes on CPU is reached after only 2.5s by the proposed method. The number of evaluation is reduced respectively by a factor 95 and 328 compared to BPTT and Adjoint and computation times are divided by a factor 50 to 450.

$\dot{\mathbf{x}}$ is Newton's notation for the time derivative $\frac{d\mathbf{x}}{dt}$. $f$ and $\mathbf{\Theta}$ parameterize the dynamic, which can include dynamics where each component of $f$ is a polynomial of the state, as well as Neural ODEs as

NeurIPS 2023 AI for Science Workshop.

Table 1: Efficiency comparison: Function Evaluations (Forward/Backpropagation) and memory. Our method is more effective computationally at the cost of a controlled memory overhead. An explicit Runge-Kutta method of order $K$ uses $N(K,T) = \left(\mathcal{O}(\epsilon^{-1/K}T)\right)$ steps, our method uses $\tilde{N} = \left(\mathcal{O}(\epsilon^{-1/2K-1}T)\right)$.

| METHOD | ADAPT. | STIFF | #NFE | MEMORY | ACCURACY FWD/BWD | REF |
|---|---|---|---|---|---|---|
| ADJOINT-RK | ✓ | ✗ | $4KN$ | $\mathcal{O}(n)$ | ↑ \| ↓ | CHEN ET AL. [2018] |
| BPTT-RK | ✓ | ✓ | $2KN$ | $\mathcal{O}(nN)$ | ↑ \| ↑ | GRUSLYS ET AL. [2016] |
| ACA-CVODE | ✓ | ✓ | $4KN$ | $\mathcal{O}(nN)$ | ↑ \| ↑ | ZHUANG ET AL. [2020] KIM ET AL. [2021] |
| LTC | ✗ | ✓ | $2KN(4,T)$ | $\mathcal{O}(nN(4,T))$ | ↑ \| ↑ | HASANI ET AL. [2020] |
| THIS PAPER | ✗ | ✓ | $2K\tilde{N}$ | $\mathcal{O}(nK)$ | ↑ \| ↑ | |

in Chen et al. [2018]. Moreover, it is not necessary for observations to be simultaneously available for each dimension; the loss is only computed for the dimensions where observations are present.

## 2 Background and related work

The inverse problem (1), has been widely studied in optimal control where it is referred to as system identification and in scientific machine learning, with renewed interest since the Neural ODEs in Chen et al. [2018], Rubanova et al. [2019]. The existing literature splits into direct approaches that solve Problem (1) using numerical integration method (ODE Solvers), and surrogate methods.

**Direct approaches:** In this family of methods, the continuous Problem (1) is solved by alternating between numerical integrations of the dynamic given a choice of parameters $\Theta$ and gradient-based parameter updates. Balancing accuracy, memory, and computational complexity leads to a variety of techniques. The continous version of the Backpropagation Through Time (BPTT) method used to train Recurrent Neural Networks (RNNs) is obtained by differentiating through the operations of an ODE Solver using an automatic differentiation framework such as Pytorch Paszke et al. [2019] or JAX Bradbury et al. [2018], or through a custom ODE Solver as for Liquid Time-Constant (LTC) networks in Hasani et al. [2020] or in Forgione and Piga [2021]. Memory usage can be decreased compared to these methods by utilizing one additional backward numerical integration to estimate the gradient using the Pontryagin principle in the adjoint sensitivity method, as described in Chen et al. [2018]. To solve numerical discrepancies between the forward and backward integrations, adaptive checkpointing (ACA) methods have been developed in Zhuang et al. [2020] to partially mitigate the problem, while in Kim et al. [2021] the forward pass is completely stored in memory and only the adjoint is integrated backwards. Inherently, direct methods require an estimate of the initial condition, which can reduce the accuracy of the gradients on chaotic systems so that shooting methods can be used to consider smaller subtrajectories, albeit reducing the ability of these methods to interpolate between observations. We compare the memory and computational complexity of the method in this paper to direct methods in Table 1. The initialization of parameters may impact convergence as the optimization landscape is non convex, see Varah [1982]. Additionally, bifurcations may lead to instability, where random weight initialization or updates can cause trajectories to diverge or require numerous adaptive steps. See Section 4 for a comparison of wall-clock times among methods.

**Surrogate methods:** Gradient matching was proposed in Varah [1982] and involves smoothing the trajectory to estimate the trajectory and its time derivative (the gradient) before adjusting parameters $\Theta$ to match the estimated gradients along the trajectory. Different optimization algorithms are discussed in Varah [1982], Ramsay et al. [2007], Tjoa and Biegler [1991], which involve block descent and various update techniques. Niu et al. [2016] provides a framework for these algorithms by using a reproducing kernel approach. The Sparse Identification of Nonlinear Dynamics (SINDy) framework, introduced in Brunton et al. [2016] combines gradient matching with sparse regression when $f$ is a linear combination of nonlinear functions. Weak formulations and integral form using trapezoidal integration for regularly sampled data are presented in Messenger and Bortz [2021], Schaeffer and McCalla [2017]. Calderhead et al. [2008], Dondelinger et al. [2013] have explored Bayesian approaches that combine gradient matching with sampling strategies and Bayesian updates.

## 2.1 Collocation methods

In recent decades, the use of collocation methods for computing numerical solutions to optimal control problems has become increasingly popular (see Betts [2010], Ljung [1999]). These are implicit integration techniques in which the values of the state and control (in our case, the parameters $\Theta$) at the discretization nodes are treated as decision variables. Constraints ensure that the dynamics are satisfied at the nodes, resulting in a nonlinear optimization problem. Although, as noted in Varah [1982], these methods justify gradient matching methods, to the best of our knowledge, their use as numerical integration method to the learning problem is original. In this paper, we use the selection of nodes from the Legendre-Gauss-Radau (LGR) method for its suitability to initial value problems. This choice results in an implicit technique that is A-stable, i.e., with numerical stability guarantees for classes of initial value problems, and symplectic, i.e., preserving the Hamiltonian of the system. Additionally, it has an approximation error of $o(h^{2K-1})$, where $K$ is the degree of the approximating polynomial and $h$, the size of the time step. see Fahroo and Ross [2008], Garg et al. [2011] for more discussions and proofs. Other choices of collocation method are compatible with our approach, although, as shown in Wei et al. [2016], using a Gaussian quadrature collocation method ensures that the KKT optimality conditions on the discretized problem are a discretization of the Pontryagin Principle, which connects our method to the adjoint sensitivity methods mentioned earlier.

## 2.2 Contributions

This paper presents a new surrogate approach addressing the direct problem on the trajectory by using collocation methods to eliminate the need for estimating derivatives from (noisy) data, and to provide accurate integral solutions without relying on ODE solvers. The method scales well with dataset size and horizon length $T$, by using low-dimensional regressions on fixed length subintervals, and to large dimensions of the state as components can be learned independently in parallel.

We evaluate our approach on synthetic experiments and also examine experimentally the limitations of our method's reliance on data and design principles. In particular, interpolation before learning coefficients can lead to failure if the available data cannot recover a relevant trajectory.

## 3 Algorithm

The algorithm, *Batched Integral Matching*, whose pseudo code is Algorithm 1, alternates between linear regressions estimating vectors $\mathbf{X}_l$ of values of the state at LGR points and gradient descents minimizing the loss $\ell$:

$$\ell(\mathbf{\Theta}) = \sum_{i=1}^{N-1} \sum_{l=1}^{n} \|\mathbf{D_K}^{-1}\mathbf{F}_i(\mathbf{X_l^k}, \mathbf{\Theta}) - \mathbf{X_{l\,i}^k}\|^2$$

Which is a collocation based approximation of:

$$\sum_{i=1}^{N-1} \sum_{l=1}^{n} \sum_{k=1}^{K} \| \int_{t_{i-1}}^{t_{i-1}+h_i\tau_k} \dot{\mathbf{x}}(t) - \mathbf{f}(\mathbf{x(t)}, \mathbf{\Theta})dt\|^2$$

Where $h_i = t_{i+1} - t_i$. The separable structure of the loss $\ell$ allows for batching strategies. Section 3.1 details the steps from the continuous problem to the problem of minimizing the introduced loss $\ell$. Section 4 presents experimental results.

---

**Algorithm 1** Batched Integral Matching

1: **Input:** data $(t_m, \mathbf{x}(t_m))_{m=1...M}$, order $K$, subinterval length $h$
2: **Build denoised set** $F = \{(t_f, \mathbf{x}_f(t_f)\}$ from $(t_m, \mathbf{x}(t_m))_{m=1,...,M}$
3: **initialize** $\Theta$
4: **repeat**
5:    **initialize** Gradient estimate $\nabla\ell_\Theta = 0$
6:    **generate** a set $S$ of subintervals $[a, a+h]$
7:    **for** $s$ **in** $S$ **do**
8:       **compute** $X_s$ with $n$ Ridge regressions on $s$, using $F$
9:       **compute** $\nabla\ell(\Theta, X_s)$, gradient wrt $\Theta$ of the loss given $X_s$
10:      **accumulate:** $\nabla\ell_\Theta += \nabla\ell(\Theta, X_s)$
11:    **end for**
12:    **update** $\Theta$: $\Theta \leftarrow$ update(step, $\frac{1}{|S|}\nabla\ell_\Theta$)
13: **until** Convergence or $maxIter$ is reached

---

### 3.1 Theoretical foundations of the algorithm

This section presents the derivation of the loss $\ell$ and its theoretical motivation.

We discretize the continuous problem (1) using a multistep LGR collocation: the state is approximated by continuous piecewise polynomials of degree $K$ on a subdivision: $0 = t_1 < \ldots < t_N = T$. Then, subintervals are rescaled to $[0, 1]$, and the polynomials are represented in the Lagrange polynomials basis $(l_j)_{j=0,...,K}$ associated with the LGR points $(\tau_j)_{j=0,...,K}$: on the $i$th subinterval, of length $h_i = t_{i+1} - t_i$, we use the change of variable: $t = t_i + \tau h_i$. Using the vector $\mathbf{X_i} \in \mathbb{R}^{n(K+1)}$ obtained

by stacking the $\mathbf{x}_{ij} = \mathbf{x}(t_i + h_i\tau_j)$ by component, then index $j$, the state and its derivative are:

$$\forall t \in [t_i, t_{i+1}], \tau = \frac{t - t_i}{h_i}, \ \mathbf{x}(t) = \sum_{j=0}^{K} l_j(\tau)\mathbf{x}_{ij} = \mathbf{V}(\tau)\mathbf{X}_i, \ \dot{\mathbf{x}}(t) = \frac{1}{h_i}\sum_{j=0}^{K} l'_j(\tau)\mathbf{x}_{ij} = \frac{1}{h_i}\mathbf{D}(\tau)\mathbf{X}_i$$

Using this approximation, we obtain the classical collocation based formulation, Problem (2):

$$\min_{\substack{\mathbf{\Theta}, \mathbf{x}_0 \\ (\mathbf{X}_i)_i}} \quad \frac{1}{M}\sum_{i=1}^{N-1}\sum_{\substack{m\in[M] \\ t_m\in[t_i,t_{i+1}]}} \left\| \mathbf{V}\left(\frac{t_m - t_i}{h_i}\right)\mathbf{X}_i - \mathbf{x}(t_m)\right\|^2$$

$$\text{s.t.} \quad \mathbf{V}(0)\mathbf{X}_1 = \mathbf{x}_0, \tag{2a}$$

$$\mathbf{V}(0)\mathbf{X}_i = \mathbf{V}(1)\mathbf{X}_{i-1}, \quad i \in 2, ..., N-1, \tag{2b}$$

$$\mathbf{D}(\tau_j)\mathbf{x}_i = h_i f(\mathbf{x}_{ij}, \mathbf{\Theta}). \quad \substack{i\in[N-1], \\ j\in 1,...,K.} \tag{2c}$$

Noting that the continuity constraints (2b) do not directly link the variables $\mathbf{X}_i$ and $\mathbf{X}_j$ when $i \neq j$, their relaxation transforms the problem into a multiple subtrajectories problem with shared parameters $\mathbf{\Theta}$, with a natural link to shooting methods.

As detailed in appendix D, the single subinterval problem has a special structure that is later used to reformulate the multi-trajectories problem: on a single subinterval $[a, a + h]$, Problem (2) has a block diagonal matrix of constraints $\tilde{\mathbf{D}} = diag(\tilde{\mathbf{D}}_K, \ldots, \tilde{\mathbf{D}}_K)$, where $\tilde{\mathbf{D}}_K$ is invertible and $\mathbf{F}$, which consists of evaluations of $f$ at the collocation nodes:

$$\min_{\substack{\mathbf{\Theta}, \\ \mathbf{X}}} \quad \frac{1}{M}\sum_{m\in[M]} \left\| \mathbf{V}(\frac{t_m - a}{h})\mathbf{X} - \mathbf{x}(t_m)\right\|^2$$

$$\text{s.t.} \quad \tilde{\mathbf{D}}\mathbf{X} = \mathbf{F}(\mathbf{X}, \mathbf{\Theta}). \tag{3}$$

The $k$th component of $\mathbf{F}$ is: $\mathbf{F}(\mathbf{X}, \mathbf{\Theta})_k = \left(\mathbf{X}_{k(K+1)}, hf(\mathbf{x}(\tau_1), \mathbf{\Theta})_k, \ldots, hf(\mathbf{x}(\tau_K)), \mathbf{\Theta})_k\right)^T$.

Using the invertibility, we consider from now on the problem in integral form

$$\min_{\substack{\mathbf{\Theta}, \\ \mathbf{X}}} \quad \frac{1}{M}\sum_{m\in[M]} \left\| \mathbf{V}(\frac{t_m - a}{h})\mathbf{X} - \mathbf{x}(t_m)\right\|^2$$

$$\text{s.t.} \quad \mathbf{X} = \tilde{\mathbf{D}}^{-1}\mathbf{F}(\mathbf{X}, \mathbf{\Theta}). \tag{4}$$

Lastly, $\tilde{\mathbf{D}}_K^{-1}$ has a special structure: $\forall k, \tilde{\mathbf{D}}_K^{-1}\mathbf{F}(\mathbf{X}, \mathbf{\Theta})_k = \mathbf{X}_{k(K+1)} + h\mathbf{D}_K^{-1}\tilde{\mathbf{F}}(\mathbf{X}, \mathbf{\Theta})_k$ where $\tilde{\mathbf{F}}(\mathbf{X}, \mathbf{\Theta})_k = (f(\mathbf{x}(\tau_1), \mathbf{\Theta})_k, \ldots, f(\mathbf{x}(\tau_K)), \mathbf{\Theta})_k)^T$ and $\mathbf{D}_K^{-1}$ is another matrix that can be precomputed. Details and proofs can be found in appendices D and E and rely on permutations of constraints by component and the invertibility is due to the polynomial interpretation of the differentiation matrix $\mathbf{D}$. The special structure allows for a reduction in complexity of the matrix multiplication from a $K + 1$ by $K + 1$ square matrix to a $K$ by $K$ square matrix.

Finally, we obtain Problem (5) from Problem (2) by relaxing continuity constraints and, using augmented lagrangian ideas, by introducing a quadratic penalty on the integral form, weighted by $\rho > 0$ discussed after:

$$\min_{\substack{\mathbf{\Theta}, \\ (\mathbf{X}_i)_i}} \quad \frac{1}{M}\sum_{i=1}^{N-1}\sum_{\substack{m\in[M] \\ t_m\in[t_i,t_{i+1}]}} \left\| \mathbf{V}\left(\frac{t_m - t_i}{h_i}\right)\mathbf{X}_i - \mathbf{x}(t_m)\right\|^2 + \rho\sum_{i=1}^{N-1}\left\|\tilde{\mathbf{D}}^{-1}\mathbf{F}_i(\mathbf{X_i}, \mathbf{\Theta}) - \mathbf{X_i}\right\|^2. \tag{5}$$

Using the block diagonal structure of $\tilde{\mathbf{D}}$ and expanding the norms, the problem becomes:

$$\min_{\mathbf{\Theta}} \quad \frac{1}{M}\sum_{i=1}^{N-1}\sum_{l=1}^{n}\underbrace{\sum_{\substack{m\in[M] \\ t_m\in[t_i,t_{i+1}]}} \left\| \mathbf{V}\left(\frac{t_m - t_i}{h_i}\right)\mathbf{X}_{il} - \mathbf{x}_l(t_m)\right\|^2}_{\substack{r_{il}(\mathbf{X_{il}}) \text{ least square regression} \\ \text{estimating the values at LGR nodes from data}}} + \underbrace{\rho\, M\|\tilde{\mathbf{D}}_\mathbf{K}^{-1}\mathbf{F}_i(\mathbf{X_i}, \mathbf{\Theta})_l - \mathbf{X_{il}}\|^2}_{s_i(\mathbf{X_i}) \text{ system inversion}}.$$

$$\tag{6}$$

We solve this last formulation with Algorithm (1), inspired by the Alternating Direction Method of Multipliers (ADMM) from Boyd et al. [2011]. The algorithm alternates between linear regressions that estimate the trajectory at LGR nodes and system inversion steps that use gradient descent to fit parameters $\Theta$ to match integrals. We set $\rho = 1$ as it only affects gradient estimation, not parameter optimization. In practice, good results can be obtained by using a pool of overlapping subintervals and going over the pool during training epochs, thus caching computations from linear regressions. Note: using a pure ADMM approach would have brought the algorithm close to the alternating updates of parameters of an interpolating Reproducing Kernel and the parameters of ODEs to be infered, as studied in Niu et al. [2016]. In that latter case, and more broadly, the use collocation methods in an integral form leads to a problem on the trajectory directly, which is generally easier to estimate than the derivative. Such a difference may explain the enhancement in robustness to noise on observations over gradient matching methods as observed in section 4.

The design of the loss is such that, if the estimates at Gauss-Radau nodes are accurate and the loss at convergence is zero, then the simulation of the trained model will be an exact match to the observed trajectory of the system, apart from the integration error due to the collocation method of order $K$.

**Reexamining the relaxation of continuity**  Relaxing continuity between adjacent subintervals controls the propagation of integration errors over the horizon, and also makes the method less vulnerable to inaccurate initial conditions. Those two properties are to learn chaotic systems. Ultimately, overlapping intervals allow the continuity of state components contained in the data to be transferred back to the estimated trajectory by data and ultimately to the solution through the loss function.

**Speed-ups**: Evaluating $\tilde{\mathbf{F}}$ and its gradient requires the evaluation of $f$ at each LGR node of each subinterval. Since the values at these nodes are already known when estimating the gradient, the evaluation of the value and the gradient is performed in the software and hardware optimized setting of a parallel batch evaluation. Furthermore, when parameters $\Theta$ can be partitioned by state component, such as in polynomial dynamics, problem (6) separates by component that can be learned independently and in parallel, facilitating the learning of high-dimensional states.

**Estimating the trajectory: the added value of denoising**  The performance of the proposed method is dependent on the accuracy of the values estimated at the collocation nodes. The distribution of discretization nodes over the interval $[0, 1]$ is not uniform, with more density near the boundaries and can lead to artifacts in noisy settings at the boundaries. To address this, we use a simple sliding window technique inspired by Savitzky and Golay [1964]. See Appendix A for more details.

## 4    Experiments

For each dynamical system in the benchmark, each experiment consists in running a simulation of the dynamic from a random initial condition and running the algorithms on observations corrupted by with Gaussian noise. As our method is capable of handling arbitrary dynamics, we use baselines that are suited to structure of $f$. We also report runtimes and compare the number of function evaluations.

We study the raw performance of the training procedure on Section 4.1, higher-dimensional problems in Section 4.2, failure modes of the algorithm in Section 4.3 and its complexity in Section 4.4.

### 4.1    Raw performance on the learning of accurate models from noisy observations

We first consider the following canonical examples of chaotic systems: the Lorenz 63 attractor Lorenz [1963], the Rossler attractor Rössler [1976], the Duffing model Duffing and Emde [1918]. Those systems are of dimensions up to 4 and are polynomials of degree up to 3. We implemented the algorithm in Python (PyTorch and JAX) and Julia.

**Learning Polynomial dynamics:**  We start by fitting the coefficients of polynomial dynamics of degree 3 that contain the original equations along with other terms and compare our method (IMATCH) to the SINDy approach from Brunton et al. [2016], for different levels of noise. These experiments, reported in Table 2, show that our algorithm learns meaningful models, even when the observations are noisy. It should be noted that no regularization was used in our method for these experiments. SINDy, on the other hand, applies thresholding to coefficients which aids its accuracy, as the underlying model is sparse and could explain why SINDy has a slight advantage in noiseless cases. In noisy conditions, our method, which operates on a smoothed trajectory rather than SINDy's finite differences, has a considerable edge.

Table 2: On polynomial dynamics: for $T = 40$, our method with subintervals of length 1 and an integration order $K = 30$ outperforms the SINDy method in noisy settings

| | | | | NOISE | | |
|---|---|---|---|---|---|---|
| METRIC | MODEL | METHOD | 0% | 5% | 10% | 20% |
| RMSE (%) $\dot{\mathbf{x}}$(%) | LORENZ63 | SINDY | **0.18** | $7.45 \pm 5.9$ | $10.77 \pm 0.4$ | $22.95 \pm 3.4$ |
| | LORENZ63 | IMATCH | 0.25 | $\mathbf{1.62 \pm 1.4}$ | $\mathbf{4.63 \pm 3.6}$ | $\mathbf{8.81 \pm 4.3}$ |
| | ROSSLER | SINDY | 0.02 | $4.3 \pm 0.5$ | $12.15 \pm 1.0$ | $26.95 \pm 2.3$ |
| | ROSSLER | IMATCH | $< \mathbf{10^{-2}}$ | $\mathbf{0.46 \pm 0.1}$ | $\mathbf{0.92 \pm 0.2}$ | $\mathbf{2.11 \pm 0.5}$ |
| | DUFFING | SINDY | **0.01** | $5.64 \pm 0.3$ | $9.79 \pm 0.3$ | $13.27 \pm 1.2$ |
| | DUFFING | IMATCH | 0.34 | $\mathbf{2.47 \pm 1.6}$ | $\mathbf{4.18 \pm 2.2}$ | $\mathbf{8.9 \pm 3.8}$ |

Table 3: We conducted a benchmark of 40 runs of each algorithm on a CPU to compare the runtime performances of learning a spiral dynamic in Chen et al. [2018] using a neural network with one hidden layer containing 50 neurons. Our algorithm was found to compute gradients almost two orders of magnitude faster than the backpropagation through the solver (BPTT) and three orders of magnitude faster than the adjoint. Computation times were measured for 2000 iterations of the adjoint/BPTT with the default parameters from the official Neural ODE library and a stopping criterion on the loss of $10^{-4}$ for our method. The results are given as the 1st and 9th decile intervals.

| METHOD | RMSE $\dot{\mathbf{x}}$ (%) | GRADIENT ESTIMATION TIME (S) | # NFEs $(10^6)$ | TOTAL TIME (S) | SPEED-UP PER GRADIENT |
|---|---|---|---|---|---|
| OURS | $1.61 \, [1.47, 1.77]$ | $1.8 \, 10^{-5}$ | 3.1 | $2.08 \, [1.01, 3.69]$ | |
| BPTT | $1.78 \, [1.19, 3.37]$ | $5 \, 10^{-2}$ | 3.9 | $100.5 \, [88.9, 116.3]$ | 2777 |
| ADJOINT | $1.83 \, [1.15, 2.57]$ | $4.5 \, 10^{-1}$ | 28.8 | $959.8 \, [909., 993.9]$ | 25000 |

**Learning Neural ODES**  We first considered the same damped oscillator as in Chen et al. [2018] and a simple network with one hidden layer of size 50 (202 parameters) and ReLU activation, as indicated in Table 3. On this task, our method outperformed ODE Solver-based approaches by almost two orders of magnitude. To analyze the reasons for this improvement, we compared various metrics: global wall clock time, time to evaluate a gradient for a batch of 20 observations, and the number of function evaluations. In Figure 1, we present a comparison on a training trajectory with the same data and initialization strategy. The $x$ axis is the number of function evaluations. While the figure represents one training trajectory, the observations and orders of magnitude are consistent across multiple experiments. In total, our method achieves similar RMSEs on the test set with 95 to 320 times fewer function evaluations up to 99.7% reduction compared to ODE Solver methods, and, function evaluations are faster by order of magnitudes due to batching and parallelism, up to a 25,000 speed up on gradient estimation for our instance, on CPU. We then trained a simple ResNet architecture with 300 hidden units (90,000 parameters) and two shared residual blocks with ReLU activations, on the Lorenz63 model and present results in Table 4. On a Tesla T4 GPU, an RMSE of around 1% was obtained within 20 minutes with our method. See appendix B for more details.

**Learning coefficients in nonlinear structures**  The FitzHugh–Nagumo model FitzHugh [1961] is a common benchmark in the gradient matching literature as though this model is polynomial in the state, the parameters to be inferred have nonlinearities contrary to the previous systems. We find that our method yields similar errors on the coefficients as the Bayesian approach in Calderhead et al. [2008] in only 5s, which is at least two times faster than the results from this reference.

Table 4: Learning the Lorenz63 model with Neural Networks: comparison of performance between our method and adjoint based methods for various execution times, 5% noise added.

| METHOD | RMSE $\dot{\mathbf{x}}$ (%) 3 MIN. | RMSE $\dot{\mathbf{x}}$ (%) 5 MIN. | RMSE $\dot{\mathbf{x}}$ (%) 10 MIN. |
|---|---|---|---|
| INTEGRAL MATCHING | $4.31 \pm 0.5$ | $3.8 \pm 0.4$ | $3.4 \pm 0.4$ |
| BPTT | $41.51 \pm 3.29$ | $32.20 \pm 2.46$ | $21.83 \pm 2.13$ |
| ADJOINT | $91.87 \pm 2.93$ | $87.89 \pm 4.37$ | $80.73 \pm 4.79$ |

## 4.2 Learning chaotic systems in high dimensions

The Lorenz [1996] family of models represent chaotic systems of arbitary dimensions. We focus on the $40$ dimension model with forcing terms of $16$, exhibiting 9 positive eigenvalues for the linearized equation around the equilibrium, leading to very chaotic evolutions, see Sapsis and Majda [2013].

This model represents a significant jump in complexity compared to the previous canonical examples. We consider trajectories obtained from an initial equilibrium with a disturbance that is typically below the numerical tolerance errors: these systems can fail many adjoint methods from the start. To promote sparsity, we used a simple sequential thresholding heuristic: small values of parameters were successively projected to $0$ and the model retrained. The results presented in Figure 2 exhibit a phase transition that we conjecture to be associated with sparsity and the large dimension of the system.

## 4.3 Limitations and impact of data quality and availability on the learned model

Taking an opposite view to observing longer trajectories, we observe a phase transition to failure when the available signal decreases: the RMSE of the learned model increases with the signal-to-noise ratio until a certain point where the algorithm's performance deteriorates. Denoising delays the performance collapse and slows down error growth but when the interpolation is not relevant for too many subintervals to be averaged, convergence fails.

This hints at a limitation of the presented version, which relies on fixed length subintervals and orders. If the integration order $K$ needs not be fixed (aside from easing implementation and providing a rigid computation graph), we can in principle use variable-length subintervals. Another observed failure mode on neural ODEs happens when the integrated trajectory diverges from the trajectory that was fitted: contrary to adjoint based methods that have been trained around the training trajectory, potentially outside of

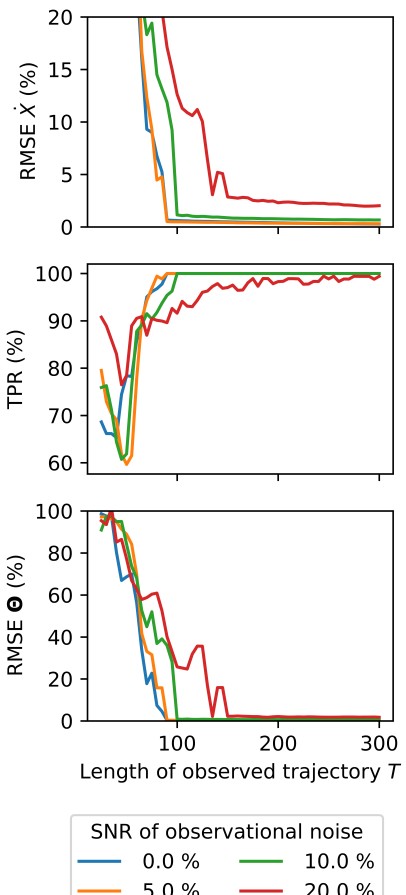

Figure 2: For the Lorenz96 model, for various levels of noise and observed length, we plot (top) The RMSE of the error on the derivative, (middle) the true positive rate of the non zero terms identified in equations, (bottom) the relative error on parameters

its manifold, our method is sharp around the observations, so that the structure of equations or the variety of the observed trajectory is paramount. To solve these limitations, given its speed, our method can be used to warm start an ODE Solver method, with a relevant prior.

## 4.4 Comparison of the number of backpropagations with ODE solvers methods

We complement the theoretical estimates from Table 1 by a experimental comparison on the Lorenz63 model. While adaptive methods involve varying step numbers during descent, our analysis provides a static estimation that hints at the significant computational gap between methods. To avoid interference with measurement times, we only compute the state at $t = T$ ($M = 1$). We present averaged results for ODE solvers recommended for Neural ODEs in Table 5. Experiments reveal that

our method requires between 10 to 40 times fewer backpropagations on the neural network $f$ than standard stiff methods for Neural ODEs. On the Lorenz96 with 40 dimensions, this gap widens further to a factor of 20 when comparing $K = 30$ to its closest contender BDF. Looking at Table 5 and Figure 3, a natural question is the choice of $K$. Increasing $K$ improves accuracy but also increases data requirements and computational costs due to the super-quadratic complexity of matrix multiplication (Strassen or Fawzi et al. [2022]). A higher $K$ might also capture more noise, putting an emphasis on denoising. In our experiments, using $K = 30$ and $h = 1$ yielded satisfactory results in terms of accuracy and runtime.

Table 5: Number of Evaluations for Different Integration Error Tolerances and $T = 1$. Our method achieves at least one order of magnitude fewer evaluations than recommended methods. Note: Dormand Prince (DoPri) results are included for reference, although it is not recommended for learning Stiff equations. Radau and BDF are the recommended methods.

| | ABS. TOLERANCE | | |
| METHOD | $10^{-3}$ | $10^{-6}$ | $10^{-8}$ |
| --- | --- | --- | --- |
| DORMAND PRINCE | $100 \pm 11$ | $199 \pm 28$ | $336 \pm 42$ |
| RADAU 5 | $148 \pm 26$ | $678 \pm 119$ | $2104 \pm 385$ |
| BDF | $95 \pm 16$ | $274 \pm 62$ | $649 \pm 113$ |
| LGR, $K = 5$ | 27 | 142 | 333 |
| LGR, $K = 8$ | 22 | 72 | 133 |
| LGR, $K = 20$ | 10 | 50 | 71 |
| LGR, $K = 30$ | 6 | 50 | 59 |

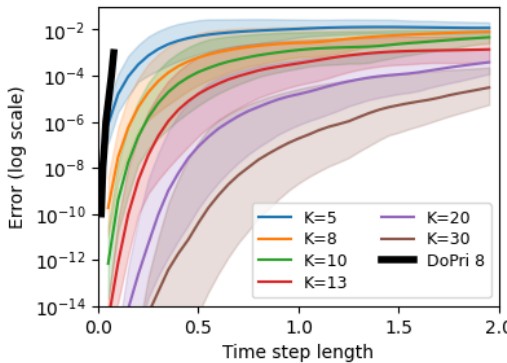

Figure 3: Integration error (avg.) on Lorenz63 vs. step length for DoPri and various orders.

## 5    Conclusion

In conclusion, we have studied the utilization of collocation methods for system identification of nonlinear dynamical systems. While collocation methods are typically computationally intensive due to their implicit nature, we have leveraged data to simplify computations, opening up new possibilities, particularly in applications where abundant data is available and trajectories can be reliably estimated. Our method are more efficient as they require less backpropagation that existing methods to evaluate gradients at each step of the descent, and offer effective batching strategies to parallelize computations. We hope these advancements can have significant implications for the field of dynamical system modeling and help improve our understanding of underlying dynamics.

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

## A    Denoising

To denoise observations, we select a window length and degree $K$, and then perform polynomial regression to obtain denoised values in the center of the window, and, using a sliding window mechanism, constitute a set of denoised values which is then used to estimate the trajectory at LGR nodes.

The experimental results, in Figure 4, consistently show that, without filtering, the RMSE of the estimation error at the LGR nodes is higher than the error in the observations. However, filtering significantly reduces the noise in the estimates, by approximately half to one-third compared to the original observations.

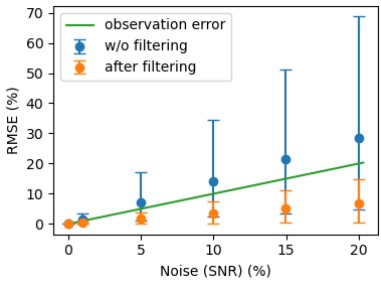 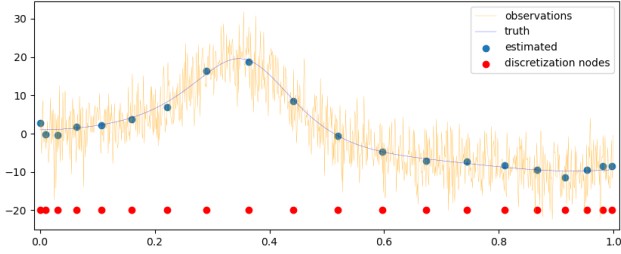

Figure 4: Denoising Performance: without filtering is worse than data. Filtering provides better estimate.

Figure 5: Observations corrupted by 40% noise. Without filtering, Estimates (blue) are irrelevant at boundaries due to uneven distribution of abscissae in the LGR nodes (red dots).

## B    Training Neural ODEs using Integral Matching

Contrary to library approaches such as in Brunton et al. [2016], Neural Networks bring less prior structure to the latent dynamic. As such, the true complexity of the manifold to learn and its translation as a data requirement especially the required length of observation to visit in different areas of the manifold is of utmost importance. Such characteristics are obviously problem specific, but there are connections with many areas studied in the physical context of finding either architectures that preserve physical quantities Raissi et al. [2017a,b] or promoting this through terms in the objective function. Promoting structure and respect of invariants brings structure to the parameters and reduces the complexity of the learning. All in all, our approach is perfectly compatible with such techniques, promoting invariants, and the loss function promotes the conservation of the Hamiltonian, though not enforcing it using projections as in Greydanus et al. [2019].

Similarly as for polynomial dynamics in section C, ie problems with more prior structure, a phase transition is observed and is linked to the architecture of the Network. Given a network with enough representative power to capture the dynamic, the phase transition is observed with regards to the availability of available data. There are several regimes, aside from terminal convergence to a relevant model that is observed in the following section where the algorithm is used to recover parameters of a dynamic within a class that contains the ground truth dynamic. We observed namely insufficient representative power and insufficient data to train the given architecture.

## C    Focus of the Lorenz 96

Contrary to the experiments in the core paper that contained no method to promote sparsity, aside from the implicit regularization of gradient descent and a small $\ell_1$ penalty that helped convergence, we used in these section a simple sequential thresholding heuristic: the problem was solved, then small values projected to $0$, then retrained on the subset of nonzero values, then thresholded again. Results are presented by Figure 6 for forcing terms $F = 8$, $F = 16$ and $F = 32$. We also provide a specific focus on $F = 8$ in Figure 7.

Better methods have been developed to recover sparse equations than the simple sequential heuristic, in a a sparse regression setting close to ours, but the fact that such a simple methods works well illustrates the interest of the loss and overall procedure. It should be noted, lastly, that such a

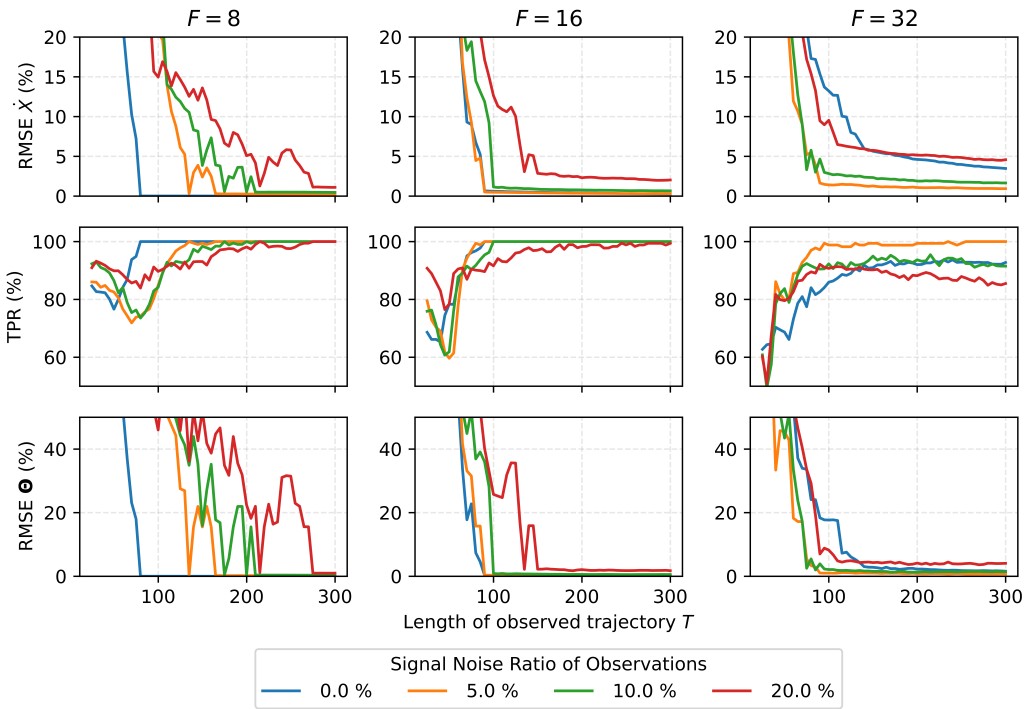

Figure 6: Phase transition: our algorithm converges to the ground truth model when given enough signal, consistently across different noise regimes. Columns: For increasing values of the forcing term $F$, corresponding to increasing chaoticity and difficulty, longer trajectories are required to recover the true dynamic. The first row indicates th RMSE of the time derivative, the second row, the true positive rate of nonzero terms recovered, the last row indicates the error on coefficients. The phase transition happens on the three metric, highlighting that past an amount of signal, our algorithm learnt the ground truth model. For $F = 8$, there is a phase transition, around $T = 80$ in the noiseless regime (blue) where the model is perfectly recovered. The greater the amount of noise, the later the phase transition. Interestingly, we observe that asymptotically on $T$, our model converges to a relevant solution for various noise levels, ie the red curve with the noise of $20\%$ converge to an error that keeps on decreasing with additional data. This is clearer in the second and third columns where the problem is more complex and the learning longer. After a phase transition, happening around $T = 100$, even $T = 130$ for $20\%$ noise, the performance keeps on improving. On $F = 32$, another surprising phenomenon appeared: on chaotic systems, mild levels of noise appear to be beneficial to help convergence. This is possibly linked to the sparsifying heuristic being suboptimal.

computationally cheap sparsifying method is scalable to large dimensions. However, on large dimensions, for instance, $N = 200$, the number of monomials grows to more than 20,000 terms for a polynomial of degree 2, so that the method of postulating a library is doomed in higher dimensions. In such dimensions, the regression step to estimate the value at LGR nodes and the denoising process are no longer cheap, though easily treated in parallel.

However, using our method can provide an interesting option as the sequential thresholding can be used to filter and select a lower dimension set of features, so that the method speeds up as it converges to a sparser model. This points towards future work at the intersection of interpretability and sparsity on Neural ODEs.

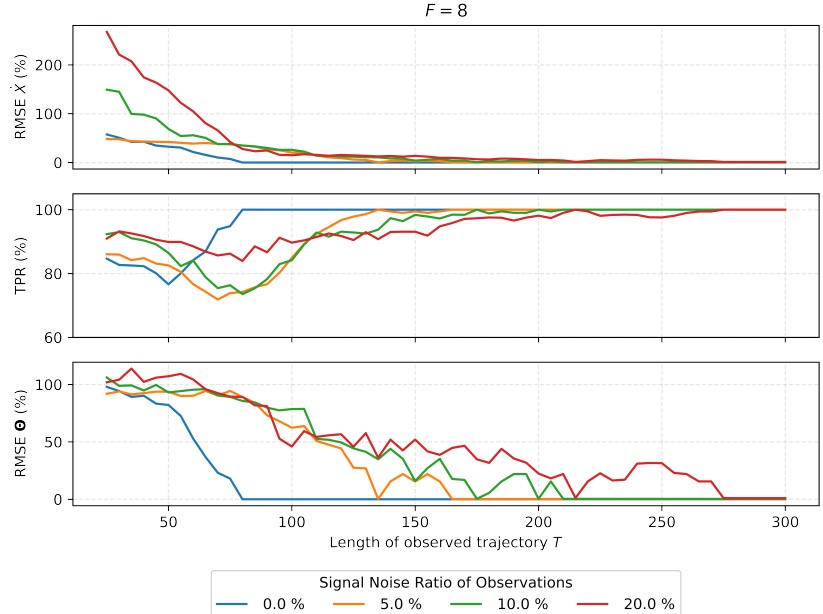

Figure 7: On the problem with $F = 8$, we present a more global view of the first column. The oscillations observed for the various metrics are likely explained by the suboptimality of the sparsifying heuristic and thresholding effects.

## D Reformulation of the single subinterval problem

The single subinterval problem has the following form:

$$\min_{\boldsymbol{\Theta}, \mathbf{X}} \quad \frac{1}{M} \sum_{m \in [M]} \left\| \mathbf{V}\left(\frac{t_m - a}{h}\right) \mathbf{X} - \mathbf{x}(t_m) \right\|^2$$

$$\text{s.t.} \qquad \mathbf{x}_0 = \mathbf{x}_0, \tag{7a}$$

$$\mathbf{D}(\tau_j)\mathbf{x} = h f(\mathbf{x}_j, \boldsymbol{\Theta}) \quad j \in 1...K. \tag{7b}$$

We reorder the constraints by components of the state first then time. By design of the collocation method, each dimension is interpolated separately. As such, grouping terms of each dimension separately, constraints naturally separate by dimension: introducing a $(K + 1) \times (K + 1)$ matrix $\mathbf{D}_K$, the constraints on the $k$th component of the state are:

$$\underbrace{\begin{pmatrix} 1 & 0 & ... & 0 \\ \dot{l}_0(\tau_1) & \dot{l}_1(\tau_1) & ... & \dot{l}_K(\tau_1) \\ \vdots & \vdots & \ddots & \vdots \\ \dot{l}_0(\tau_K) & \dot{l}_1(\tau_K) & ... & \dot{l}_K(\tau_K) \end{pmatrix}}_{\mathbf{D}_K} \cdot \underbrace{\begin{pmatrix} X_{k(K+1)} \\ X_{k(K+1)+1} \\ \vdots \\ X_{k(K+1)} \end{pmatrix}}_{X[kK:k(K+1)]}$$

$$= \underbrace{\begin{pmatrix} X_{k(K+1)} \\ h f(\mathbf{x}_1, \boldsymbol{\Theta})_k \\ \vdots \\ h f(\mathbf{x}_K, \boldsymbol{\Theta})_k \end{pmatrix}}_{F(\mathbf{X}, \boldsymbol{\Theta})_k}$$

Where $\mathbf{X}[k(K+1) : (k+1)(K+1)]$ is the projection of $\mathbf{X}$ on the span of $\{\mathbf{e}_{k(K+1)}, \dots \mathbf{e}_{(k+1)(K+1)}\}$. As the collocation is of the same order for each dimension, the matrix $D_K$ which only depends on the collocation order is the same for each dimension, so that, stacking back every component, the matrix of constraints is block diagonal:

$$
\underbrace{\begin{pmatrix} \mathbf{D}_K & & & \\ & \mathbf{D}_K & & \\ & & \ddots & \\ & & & \mathbf{D}_K \end{pmatrix}}_{\tilde{\mathbf{D}}} . \mathbf{X} = \underbrace{\begin{pmatrix} F(\mathbf{X},\mathbf{\Theta})_1 \\ F(\mathbf{X},\mathbf{\Theta})_2 \\ \vdots \\ F(\mathbf{X},\mathbf{\Theta})_n \end{pmatrix}}_{F(\mathbf{X},\mathbf{\Theta})}
$$

One final observation (proof in the following section) is that the structure of the first row of $\mathbf{D}_K$ and the collocation structure imply that the first column of the inverse of $\mathbf{D}_K$ is only composed of 1s. Namely, we have:

$$
\mathbf{D}_K^{-1} = \begin{pmatrix} 1 & 0 & \dots & 0 \\ \vdots & & \hat{\mathbf{D}} & \\ 1 & & & \end{pmatrix}
$$

This enables to compute the loss by multiplying by a $K \times K$ submatrix of $\mathbf{D}_K^{-1}$ rather than by a $(K+1) \times (K+1)$ matrix. For $K = 30$, this simple observations enables to reduce the number of operations to evaluate the product by 9% using Strassen ($O(K^2.8)$). In the end, compared to a naive implementation that would consider a product with matrix of constraint of dimension $n(K+1) \times n(K+1)$, we have transformed the problem into $n$ products with matrices of dimension $K \times K$. As the matrix is fixed, it seems from experiments that the compilation performed in JAX is able to the product for further speedups.

## E  Proof of the invertibility of matrices $\mathbf{D}_K$ and $\tilde{\mathbf{D}}$

Any element of the kernel of $\mathbf{D}_K$ can be interpreted a polynomial $P$ of degree $K$ represented in the LGR Lagrange basis. The last $K$ rows of $\mathbf{D}_K$ imply that $P$ is constant: the derivative of $P$ is a polynomial of degree $K - 1$ null at $K$ distinct points, hence null everywhere. The first row of the matrix $\mathbf{D}_K$ implies that this constant is null, ie $P = 0$. Subsequently, $\mathbf{D}$ is also invertible from its block diagonal structure of matrices $\mathbf{D}_K$. $\square$.

The first column $\mathbf{v}$ of matrix $\mathbf{D}_K^{-1}$ is a vector of ones: $\mathbf{v} = \mathbf{1}$. The first component is trivial. For the other ones, we use the adjoint matrix, algebraic manipulations and the interpretation as a differentiation matrix. Namely, have that

$$
\det(\mathbf{D}_K) = \det \begin{pmatrix} 1 & 0 & \dots & 0 \\ \dot{l}_0(\tau_1) & \dot{l}_1(\tau_1) & \dots & \dot{l}_K(\tau_1) \\ \vdots & \vdots & \ddots & \vdots \\ \dot{l}_0(\tau_K) & \dot{l}_1(\tau_K) & \dots & \dot{l}_K(\tau_K) \end{pmatrix}
$$

for any $k \neq 0$

$$
\det(\mathbf{D}_K) = \det \begin{pmatrix} \dot{l}_1(\tau_1) & \dots & \dot{l}_k(\tau_1) & \dots & \dot{l}_K(\tau_1) \\ \vdots & \vdots & \vdots & \vdots & \vdots \\ \dot{l}_1(\tau_K) & \dots & \dot{l}_k(\tau_K) & \dots & \dot{l}_K(\tau_K) \end{pmatrix}
$$

Using the adjoint matrix formula for the inverse of $\mathbf{D}_K$), to prove that $\mathbf{v} = \mathbf{1}$, we need to show that the first row of the adjoint matrix, ie the cofactors are each equal to the the determinant of $\det(\mathbf{D}_K)$. Namely, we need to show that, for any $k \neq 0, \det(A_k) = \det(\mathbf{D}_K)$ where:

$$
\det(A_k) = (-1)^k \det \begin{pmatrix} \dot{l}_0(\tau_1) & \dots & \dot{l}_{k-1}(\tau_1) & \dot{l}_{k+1}(\tau_1) & \dots & \dot{l}_K(\tau_1) \\ \vdots & \vdots & \vdots & \vdots & \vdots & \vdots \\ \dot{l}_0(\tau_K) & \dots & \dot{l}_{k-1}(\tau_K) & \dot{l}_{k+1}(\tau_K) & \dots & \dot{l}_K(\tau_K) \end{pmatrix}
$$

We form the difference $\Delta_k = \det(\mathbf{D}_K) - \det(A_k)$ and expand the determinant of $\mathbf{D}_K$ along the $k$th column, expand the determinant of $\mathbf{A}_k$ along the first column. The expansion exhibits the same

minors obtained by removing the 0th and $k$th columns. We denote $\mu_{ik}$ the determinant of the minor obtained by removing the first and $i$th row of $\mathbf{D}_K$ and the first column and $k$th column of $\mathbf{D}_K$:

$$\Delta_k = \det(\mathbf{D}_K) - \det(A_k) = \sum_{i=1}^K (-1)^{i+k} \dot{l}_k(\tau_i)\mu_{ik} - \sum_{i=1}^K (-1)^k (-1)^{i+1} \dot{l}_0(\tau_i)\mu_{ik}$$

$$= \sum_{i=1}^K (-1)^{i+k}(\dot{l}_k(\tau_i) + \dot{l}_0(\tau_i))\mu_{ik}$$

That is the difference is the determinant of a matrix $\mathbf{B}_k$

$$\Delta_k = \det(\mathbf{B}_k) = \det \begin{pmatrix} 1 & 0 & \dots & 0 & \dots & 0 \\ \dot{l}_0(\tau_1) & \dot{l}_1(\tau_1) & \dots & \dot{l}_k(\tau_1) + \dot{l}_0(\tau_1) & \dots & \dot{l}_K(\tau_1) \\ \vdots & \vdots & & \vdots & & \\ \dot{l}_0(\tau_K) & \dot{l}_1(\tau_K) & \dots & \dot{l}_k(\tau_1) + \dot{l}_0(\tau_1) & \dots & \dot{l}_K(\tau_K) \end{pmatrix}$$

Subtracting the first column from the $k$th column does not change the determinant but gives a new matrix $\tilde{\mathbf{B}}_k$ which is the same as the original matrix $\mathbf{D}_K$ except for the term on the first row and the $k$th column which equals $-1$. This matrix is not invertible: using the same interpretation as polynomials used earlier in this section, we deduce that constant polynomials ie vectors of $\mathbb{R}^{K+1}$ that are collinear to $v$ are in the kernel of this matrix.

The last $K$ rows of $\tilde{\mathbf{B}}_k v$ imply that the derivative of a constant polynomial is 0. The first row also evaluates to 0 so that $v \neq 0 \in \ker \tilde{\mathbf{B}}_k$.

Thus, $\forall k, \Delta_k = \det(\mathbf{B}_k) = 0$. $\square$

