# OpenReview forum: "Fast and Scalable Inference of Dynamical Systems via Integral Matching"
_NeurIPS.cc/2023/Workshop/AI4Science — NeurIPS2023-AI4Science Poster_

### Official Review · Reviewer_Eq8r · 2023-10-15

**Rating:** 7
**Confidence:** 3

**Review:**

In this work, the authors present a novel approach for ODE parameter identification based on collocation methods. The proposed method alternates linear regression based on LGR. Experimental results demonstrate the proposed method is more accurate and efficient comparing to baselines. In general, the paper is well-written and can be a valuable contribution to the community.

---

### Meta-Review · Area_Chair_vm8j · 2023-10-26

**Recommendation:** Accept (Poster)
**Confidence:** 3

**Metareview:**

Reviewer agrees that the paper is well fit to the workshop scope and provide a valuable contribution to the community.